# Rapid label-free analysis of *Opisthorchis viverrini* eggs in fecal specimens using confocal Raman spectroscopy

Oranat Chuchuen[1,2], Thani Thammaratana[3], Oranuch Sanpool[2,4], Rutchanee Rodpai[2,4], Wanchai Maleewong [2,4] *, Pewpan M. Intapan[2,4]

**1** Department of Chemical Engineering, Faculty of Engineering, Khon Kaen University, Khon Kaen, Thailand, **2** Research and Diagnostic Center for Emerging Infectious Diseases, Mekong Health Science Research Institute, Khon Kaen University, Khon Kaen, Thailand, **3** Research Affairs, Faculty of Medicine, Khon Kaen University, Khon Kaen, Thailand, **4** Department of Parasitology and Excellence in Medical Innovation, and Technology Research Group, Faculty of Medicine, Khon Kaen University, Khon Kaen, Thailand

* wanch_ma@kku.ac.th

## Abstract

*Opisthorchis viverrini*, a human liver fluke, is highly prevalent in Southeast Asia. Definitive diagnosis of infection is usually achieved parasitologically through the discovery of fluke eggs in feces. However, the eggs of *O. viverrini* are difficult to differentiate morphologically from those of other minute intestinal flukes in fecal samples, even for experienced technicians. The present study developed a label-free optical methodology for analysis of *O. viverrini* eggs using Raman spectroscopy. Raman features of *O. viverrini* eggs were reported that can be used as marker bands for the efficient analysis of *O. viverrini* eggs in fecal specimens. The methodology presented here allows for the rapid detection of *O. viverrini* egg infection and can be readily and practicably applied in any clinical setting, even those in which a trained parasitologist is not available.

## Introduction

Human opisthorchiasis caused by the food-borne trematode, *Opisthorchis viverrini*, remains a major public health concern, especially in Southeast Asia [1,2]. The fluke has been classified as a group 1 carcinogen [3]. In addition, chronic or recurrent infection with the parasite can cause hepatobiliary diseases such as hepatomegaly, cholangitis, cholecystitis, peri-ductal fibrosis, and gallstones and is associated with bile duct cancer (cholangiocarcinoma) [4]. Thus, early diagnosis and treatment of *Opisthorchis viverrini* infection may prevent the occurrence of these conditions. Diagnosis of opisthorchiasis is usually performed by microscopic observation of *Opisthorchis* eggs in feces. However, it is difficult to differentiate *O. viverrini* eggs from those of *Clonorchis sinensis* and the eggs of opisthorchiids from those of lecithodendriids (i.e., *Haplorchis taichui*, *Haplorchis pumilio*, and *Stellantchasmus falcatus*) and heterophyids (i.e., *Centrocestus caninus*, *Prosthodendrium molenkampi*, and *Phaneropsolus bonnei*) because of their morphological similarity [5]. In addition, microscopic examination for the presence of *O. viverrini* eggs requires specialized parasitologists working alongside well-trained laboratory

**Funding:** This research was supported by the Distinguished Research Professor Grant, Thailand Research Fund (TRF), grant no. DPG 6280002, and Khon Kaen University (PMI, WM). The contents of this report are solely the responsibility of the authors and do not necessarily represent the official views of the TRF and KKU. The funders had no role in study design, data collection and analysis, decision to publish, or preparation of the manuscript.

**Competing interests:** The authors have declared that no competing interests exist.

technicians. Thus, there is a need for a simple methodology that offers rapid and efficient analysis of *O. viverrini* eggs and does not require the presence of trained professionals, thus making detection more feasible in local hospitals and clinics in remote regions of the developing world.

Optical imaging techniques, such as Raman spectroscopy, are capable of overcoming these limitations and have emerged as powerful clinical tools. Raman spectroscopy offers promising strategies for rapid, label-free detection of small analytes and biomarkers in clinically relevant matrices. Raman spectroscopy is a vibrational spectroscopic technique that can identify molecular species based on each molecule's specific fingerprints [6]. Over the past decades, Raman spectroscopy has been widely used in the biomedical research field as a valuable clinical tool for improving clinical diagnosis. It has been shown to effective in drug concentration mapping in polymeric matrices [7,8] and tissues [9–11] and has also been used in the detection and diagnosis of a wide variety of diseases including cancers of the breast [12,13], lung [14], skin [15], prostate [16], and bladder [17].

This study aimed to investigate the feasibility of using confocal Raman spectroscopy for rapid, label-free analysis of *O. viverrini* eggs obtained from hamster and human feces. As a result, a standardized methodology for Raman spectroscopic detection of *O. viverrini* eggs was developed. Raman bands of *O. viverrini* eggs were reported, which can be used as marker bands for rapid optical analysis of *O. viverrini* eggs in fecal specimens.

## Materials and methods

### Preparation of *O. viverrini* egg specimens

Naturally infected cyprinid fish collected from a water reservoir (latitude 16.436346; longitude 102.887949, Amphur Muang, Khon Kaen Province, Thailand) were purchased from a local fish market in Khon Kaen, Thailand. No specific permission was required for sampling fishes in public locations. The dead fish were cut into small pieces, minced using a tissue homogenizer, blended with artificially digestive juice (0.25% pepsin (Sigma-Aldrich, St.Louis, MO), 1.5% HCl in 0.85% NaCl solution, NSS), and incubated in a shaking water bath at 37°C for 1 h [18]. No humane endpoint was used because only dead fish from a food market were used. The material suspension was serially filtered through 1,000-, 300-, and 106-mesh sieves. The material containing metacercariae retained on the 106-mesh sieve was washed several times with NSS through a 250-mesh sieve and collected in a sediment jar. The collected *Opisthorchis viverrini* metacercariae were examined and identified under a dissecting microscope, as previously described [1].

Five male 6- to 8-week-old Syrian golden hamsters, *Mesocricetus auratus*, average body weight 120 g, were obtained from the Animal Unit of Faculty of Medicine, Khon Kaen University. Each hamster were infected with 100 *O. viverrini* metacercariae by gastric intubation and reared in the Khon Kaen University Faculty of Medicine Animal Unit (Khon Kaen, Thailand). The study protocol was approved by the Animal Ethics Committee of Khon Kaen University (AEMDKKU 029/2018) and animal care was carried out in strict accordance with the recommendations laid out in the National Research Council of Thailand's Guide for the Care and Use of Laboratory Animals. The experiment was made to minimize pain and suffering to the animals and only fecal specimens were collected. Five male hamsters were housed under conventional conditions (12 hours dark-light cycles, temperature 25 ± 2°C), fed with a stock diet (Smart heart, Thailand) and given filtered bottled water *ad libitum*. The hamster health was observed daily and cage bedding changed twice a week. None of experimental hamsters showed serious illness or severe health problems during the study. No euthanized and killed animals were used only the fecal sediment containing *O. viverrini* eggs was collected. After

finishing this study up, the hamsters were housed for eggs collection of other studies until the animals are expired.

After 3 months of infection, a 30 mg fecal sample from each hamster was collected and suspended in 10 ml of normal saline (0.85%in distilled water), vortex and added to Mini-Parasep® (Apacor, Berkshire, England). It was then centrifuged at 2,500 rpm for 5 min. The fecal sediment containing *O. viverrini* eggs was collected in a 1.5 ml micro-centrifuge tube for examination via Raman spectrophotometry.

For *O. viverrini* eggs from infected humans, the eggs were collected from leftover stool specimens of four opisthorchiasis patients who visited Srinagarind Hospital, Faculty of Medicine, Khon Kaen University, Khon Kaen, Thailand, the endemic area of human opisthorchiasis [1]. Fresh human fecal specimens were processed using the methods for processing hamster fecal samples, as outlined above. The fecal sediment containing eggs in normal saline solution was collected for examination via Raman spectroscopy. The study protocol was approved by the Khon Kaen University Ethics Committee for Human Research (HE611407). Informed consent was obtained from adult participants and from parents or legal gardians of minors.

## Confocal Raman measurements

The configuration of the standardized experimental setup for Raman spectroscopic measurements of *O. viverrini* egg specimens is displayed in Fig 1. Briefly, 40 μL of an *O. viverrini* egg specimen was deposited onto a custom-made, quartz-bottom well (Esco Optics, Oak Ridge, NJ). Measurements were acquired with a saline buffer overlayer in order to keep the eggs in their hydrated aqueous environment during measurement using a Horiba Xplora confocal Raman microscope (Horiba Jobin Yvon, Northampton, UK) integrated with white-light video imaging. Each *O. viverrini* egg was first located though integrated white-light imaging using a 50X long-working distance objective lens (LMPLFL50X, Olympus, St. Joseph, MI) before the Raman measurements were acquired. Then, a Raman measurement of the egg was taken using an excitation source of 785 nm. Near-infrared excitation was used to minimize specimen overheating and autofluorescence. Raman spectra were recorded using LabSpec 6 software (Horiba

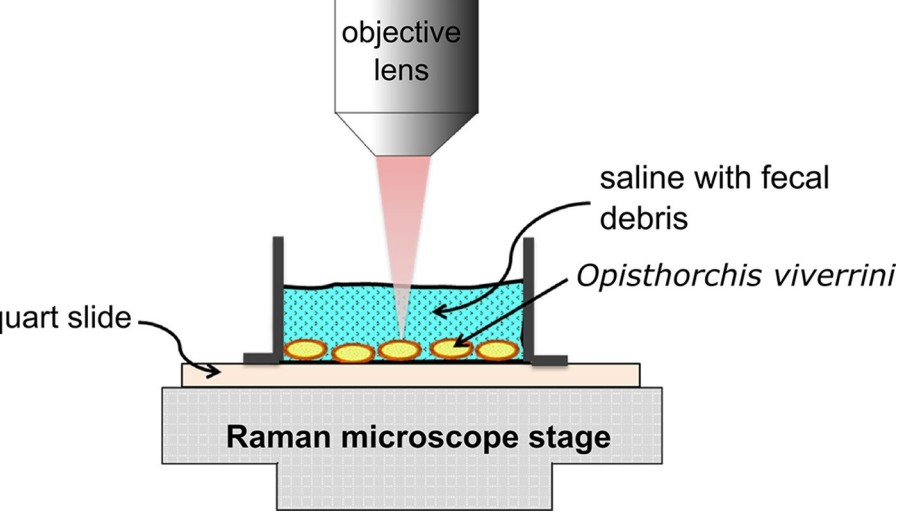

**Fig 1. Experimental setup for confocal Raman analysis of *Opisthorchis viverrini* eggs in fecal specimens.** A fecal specimen containing *O. viverrini* eggs was dropped onto a quartz-bottom well. Confocal Raman measurements of each *O. viverrini* egg were acquired while the saline overlayer remained in place.

Scientific, Edison, New Jersey) with 3 accumulations, each with an acquisition time of 60 seconds.

### Raman spectral processing

Processing of the acquired Raman spectra was conducted using LabSpec 6 software (Horiba Scientific) and MATLAB (MathWorks, Natick, MA). After each acquisition, the Raman peak locations of each acquired spectrum were immediately identified using LabSpec 6 software. Raw spectral data were also exported into Matlab for further processing. In Matlab, Raman spectra were background subtracted by a polynomial fitting routine. Next, each spectrum was normalized by its mean intensity over all wave numbers and smoothed using a Savitzky-Golay filter (12 point, degree 2).

## Results

*Opisthorchis viverrini* eggs from the fecal specimens of all infected hamsters appeared to have similar Raman spectral peaks (Fig 2). The averaged Raman spectra of 10 *O. viverrini* eggs from each infected hamster showed little variability, consisting mainly of three Raman bands. The Raman spectra of *O. viverrini* eggs from fecal specimens of four human subjects showed similar peaks as those of the infected hamsters (Figs 3 and 4). That is, *O. viverrini* eggs from feces of both hamster and human subjects possessed three major Raman peaks locating at roughly the same wavenumber. The locations of these peaks are summarized in Table 1. Minimal intra- and inter-subject variability in these Raman bands were seen, as indicated by the relatively small standard deviations in the values of the peak locations observed. This demonstrates that these characteristic Raman bands can be used as marker bands to optically detect the presence of *O. viverrini* eggs.

To confirm whether these Raman features belonged to *O. viverrini* eggs and were distinct from the Raman bands of the surrounding environment, lateral (XY-direction) and depth (Z-direction) scanning were performed. Fig 5 shows that the Raman spectra at different locations within an *O. viverrini* egg were similar and possessed the three characteristic Raman features. These characteristic Raman bands of the *O. viverrini* egg were not present outside the egg boundary (i.e., within the region around the egg containing fecal debris). Fig 6 displays a set of Raman depth scans of an individual *O. viverrini* egg from the saline overlayer into the egg and the underlying quartz slide. Within the saline overlayer, none of the spectral features of the egg was observed. The Raman features of the egg immediately appeared as the focal position moved down into the egg and disappeared again as it moved into the underlying quartz material, suggesting that the Raman bands of the *O. viverrini* egg were distinct from those of the saline and quartz material. These findings confirmed that the three Raman bands of *O. viverrini* eggs found in this study can be used effectively as marker bands for the label-free Raman analysis of *O. viverrini* eggs in fecal specimens.

## Discussion

The egg sizes and shapes of *O. viverrini* and other minute intestinal fluke are very similar and have limitations for diagnosis by using a stool examination with light microscope. Supportive tools for differentiation of *O. viverrini* eggs can be performed by the potassium permanganate staining method [1,19] and by using a scanning electron microscope [5,20]. To solve the pitfalls of the microscopic method, various sensitive and specific molecular methods have been applied such as a real-time polymerase chain reaction (PCR) assay [21] and PCR assay and pyrosequencing [22].

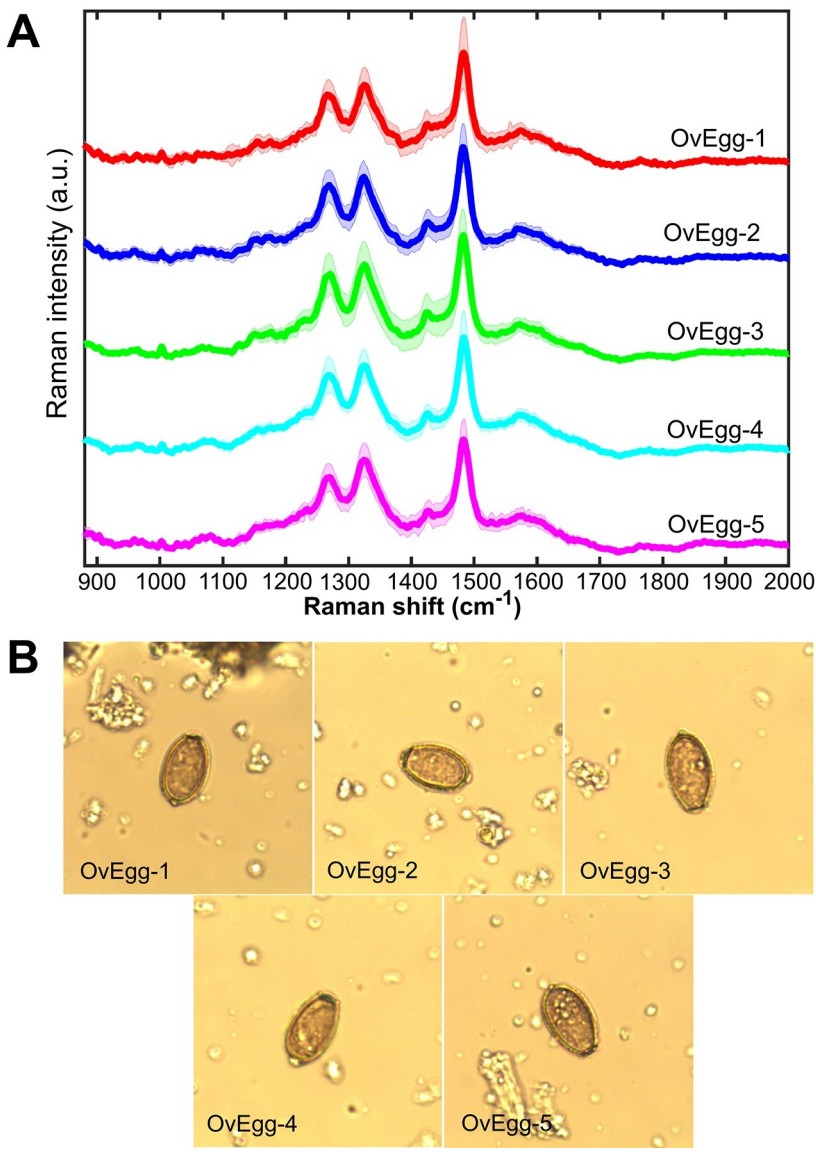

**Fig 2. Averaged Raman spectra of *O. viverrini* eggs from the feces of each infected hamster.** The averaged Raman spectra are shown (A) along with a representative microscope image of an *O. viverrini* egg from each infected hamster taken with a Raman integrated white-light imaging camera (B). The solid lines and shaded areas represent means and standard deviations of the measurements (n = 10), respectively.

In the present study, we have created and applied a label-free optical methodology for the analysis of *O. viverrini* eggs in fecal specimens using confocal Raman spectroscopy. Each individual *O. viverrini* egg from the fecal specimens of 5 infected hamsters and 4 infected human subjects possessed similar Raman vibrational bands, demonstrating minimal intra- and inter-subject variability. In addition, lateral (XY-direction) and depth (Z-direction) scanning results confirmed that these characteristic Raman bands were distinctive to *O. viverrini* eggs and were clearly distinct from the Raman bands of the surrounding sludge and the quartz material. This demonstrates that the Raman bands of *O. viverrini* eggs reported in the present study are characteristic and can be readily used as effective marker bands for the analysis of *O. viverrini* eggs in fecal environments.

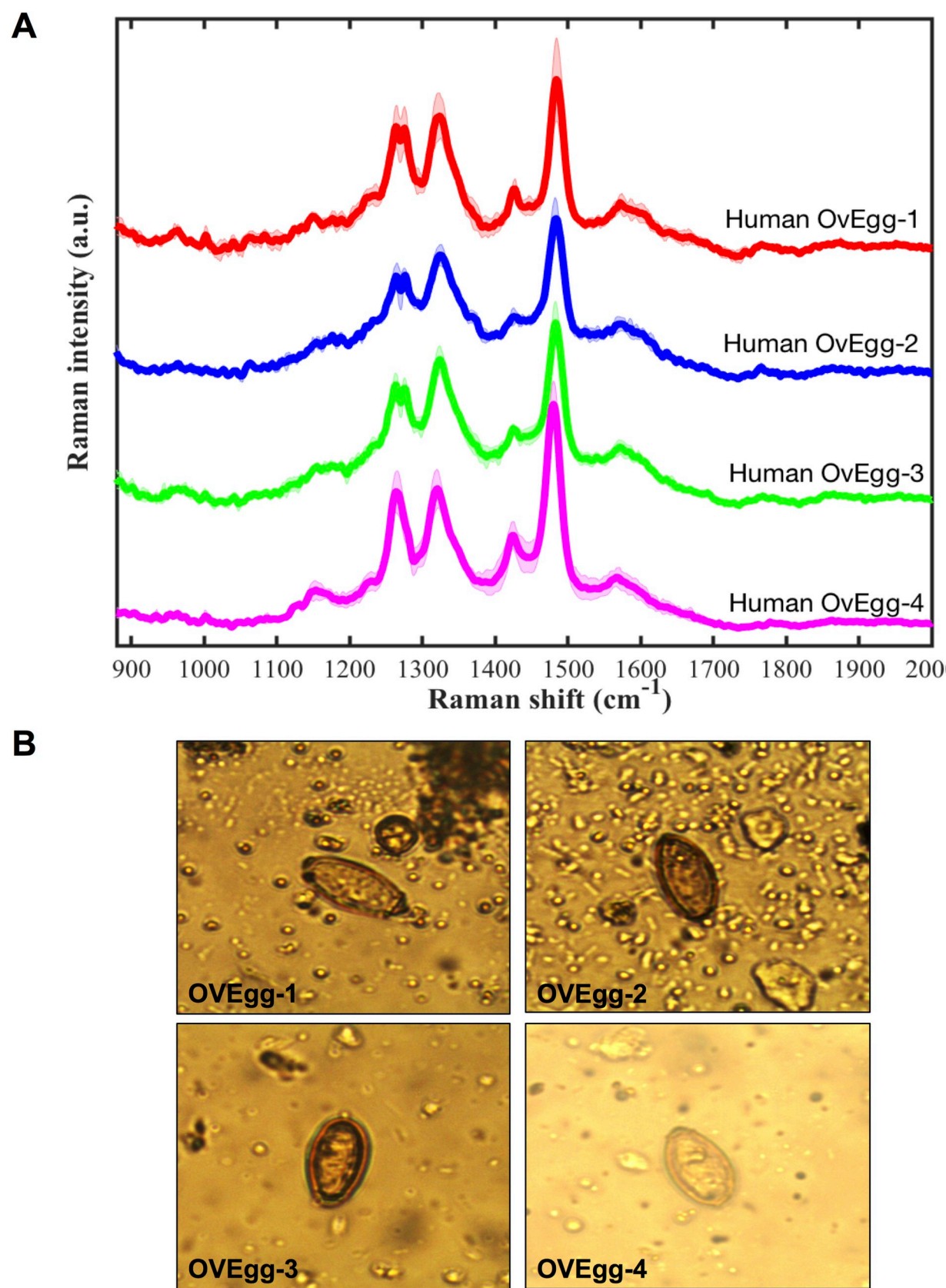

**Fig 3. Averaged Raman spectra of *O. viverrini* eggs from the feces of human subjects.** The averaged Raman spectra are shown (A) along with a representative microscope image of an *O. viverrini* egg from each patient (B). The solid lines and shaded areas represent means and standard deviations of the measurements (n = 7, 5, 6, 6 for human subjects 1 to 4, respectively).

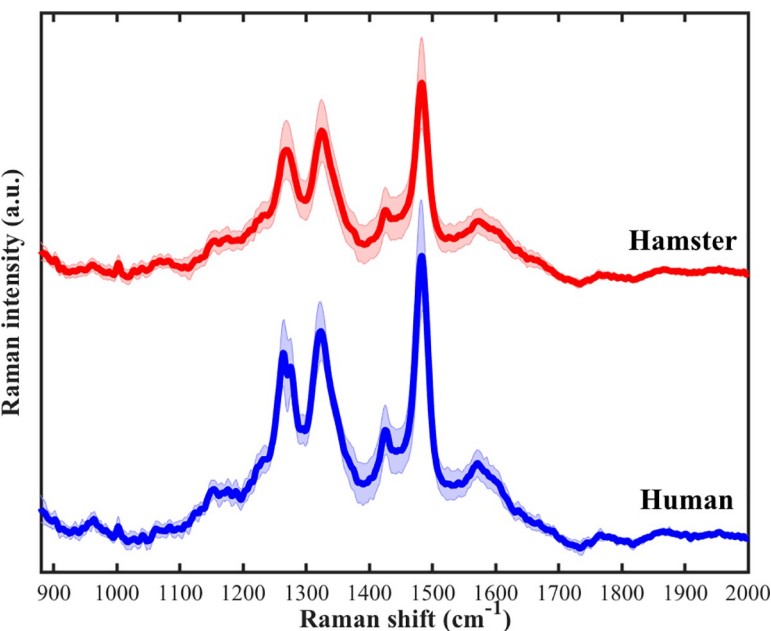

**Fig 4. Comparison of averaged Raman spectra of *O. viverrini* eggs from the feces of hamster (n = 50) and of human subjects (n = 24).** The solid lines and shaded areas represent means and standard deviations of the measurements (n = 50 for hamster and n = 24 for human).

**Table 1. Summary of characteristic Raman peak locations of *O. viverrini* eggs obtained from the fecal specimens of 5 hamsters and 4 human subjects.** Each number represents an averaged peak location ± standard deviation.

| Subject | ID | n | Raman peak locations (cm$^{-1}$) | | |
|---|---|---|---|---|---|
| | | | Peak 1 | Peak 2 | Peak 3 |
| Hamster | 1 | 10 | 1266.98±3.52 | 1326.26±2.05 | 1483.48±1.29 |
| | 2 | 10 | 1268.72±2.81 | 1324.16±2.90 | 1482.40±1.5 |
| | 3 | 10 | 1271.42±1.80 | 1325.25±1.94 | 1482.49±1.62 |
| | 4 | 10 | 1268.31±3.43 | 1325.89±2.25 | 1483.9±1.12 |
| | 5 | 10 | 1269.28±3.37 | 1325.77±2.38 | 1482.91±1.32 |
| | **Total average±SD** | **50** | **1268.94±3.27** | **1325.52±2.37** | **1483.04±1.44** |
| Human | 1 | 7 | 1267.07±5.39 | 1324.01±2.29 | 1484.61±0.51 |
| | 2 | 5 | 1265.06±0.83 | 1324.67±3.30 | 1483.21±2.27 |
| | 3 | 6 | 1264.20±1.67 | 1323.85±1.67 | 1483.17±1.40 |
| | 4 | 6 | 1264.39±1.32 | 1320.66±1.71 | 1480.14±0.79 |
| | **Total average±SD** | **24** | **1265.25±3.18** | **1323.27±2.64** | **1482.84±2.11** |

The Raman spectroscopic methodology presented is promising as a label-free technique [23] that can be used to rapidly examine *O. viverrini* eggs without the need of a well-trained professional. This can lead to a reduction in the number of well-trained microscopists required, thereby reducing overall examination time and expense. Moreover, this methodology

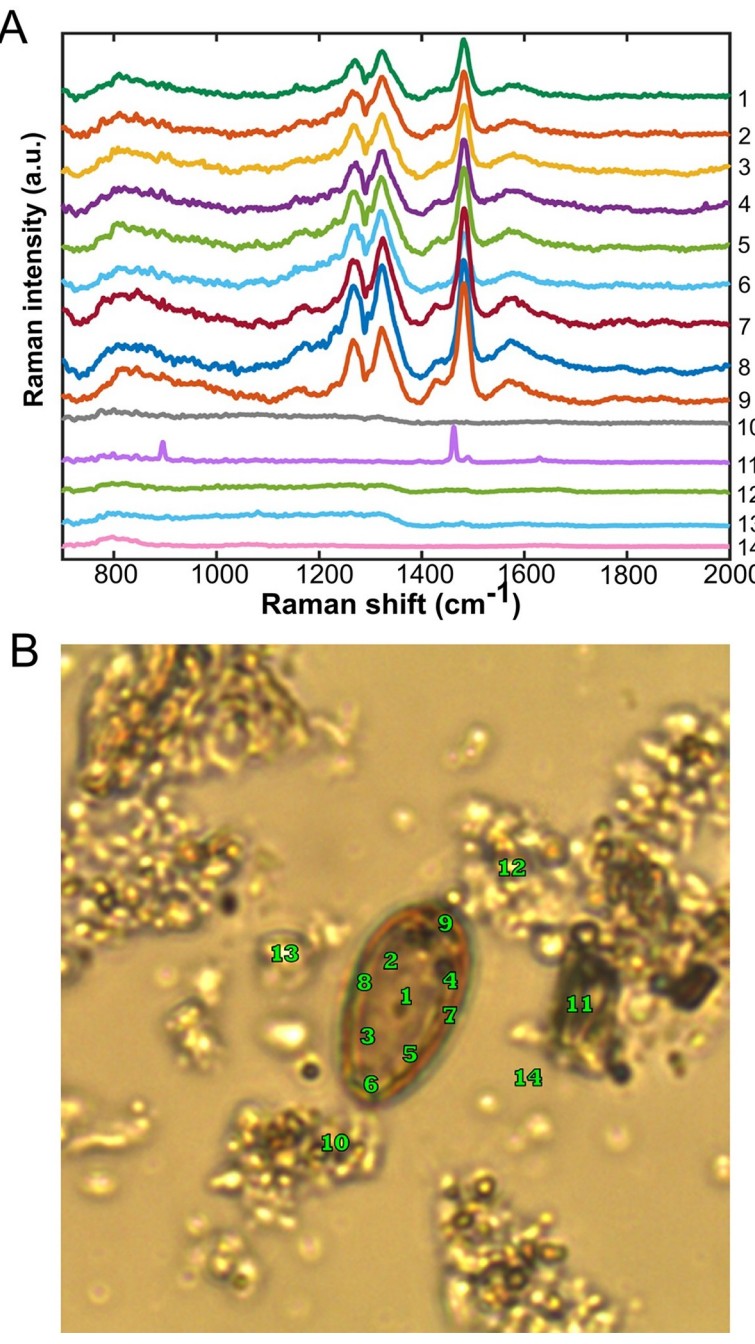

**Fig 5. Lateral Raman scanning of an individual *O. viverrini* egg.** (A) Raman spectra were taken at different spots, as shown in (B), within an *O. viverrini* egg vs. outside the egg boundary, i.e., within the region containing fecal debris. Spectra were offset for clarity of presentation.

can be extended and applied to the examination of other types of fluke eggs, including those of other minute intestinal flukes (i.e., lecithodendriid and heterophyid eggs), which are similar in both size and shape and are indistinguishable from one another though routine stool examination under a light microscope. Thus, the proof-of-principle methodology developed here could enable the analysis of different types of fluke eggs in a label-free manner, eliminating

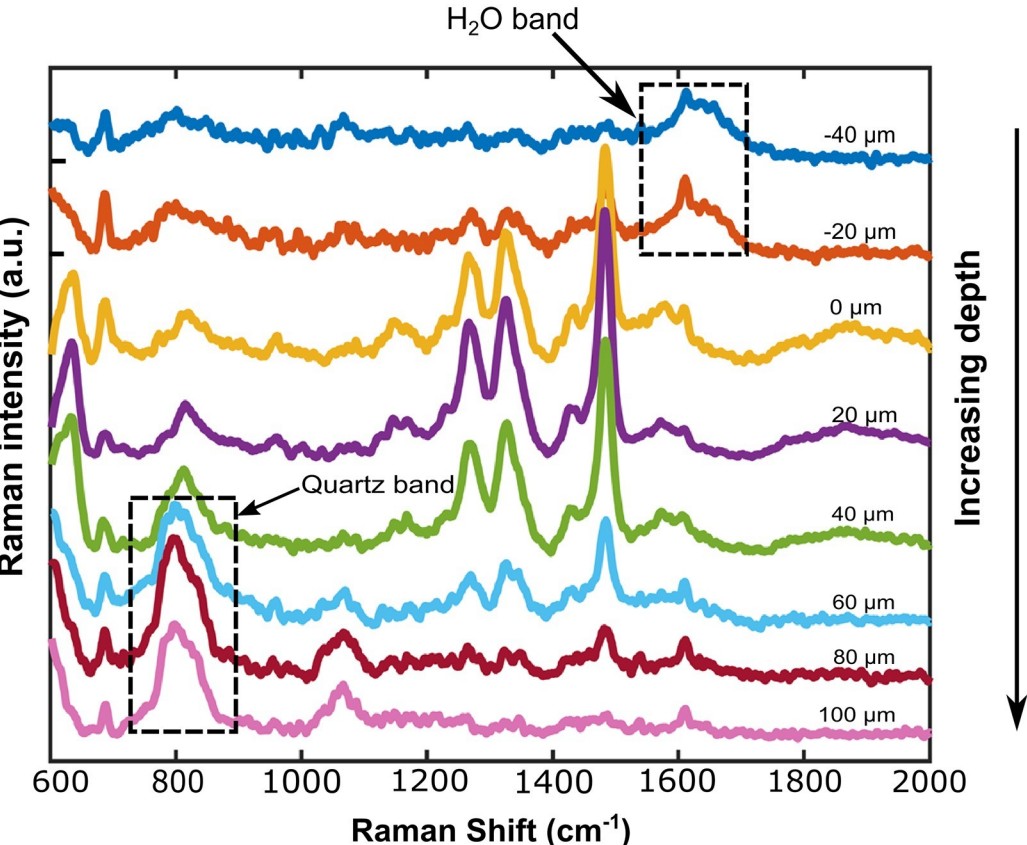

**Fig 6. Depth scanning spectra of a representative *O. viverrini* egg.** The egg was scanned from the saline overlayer through the egg and into the supporting quartz material with a 20 μm increment. Spectra were offset for clarity of presentation.

human bias and error and contributing to improvements in overall prognostic outcomes of parasitic infections.

## Acknowledgments

We wish to acknowledge the support of the English Consultation Clinic at the Khon Kaen University Faculty of Medicine Research Affairs Division and the Khon Kaen University Publication Clinic at the Research and Technology Transfer Affairs Division for their assistance in English editing.

## Author Contributions

**Conceptualization:** Oranat Chuchuen, Oranuch Sanpool, Wanchai Maleewong, Pewpan M. Intapan.

**Data curation:** Oranat Chuchuen, Thani Thammaratana, Oranuch Sanpool, Rutchanee Rodpai, Wanchai Maleewong, Pewpan M. Intapan.

**Funding acquisition:** Wanchai Maleewong, Pewpan M. Intapan.

**Investigation:** Oranat Chuchuen, Thani Thammaratana, Oranuch Sanpool, Rutchanee Rodpai, Wanchai Maleewong, Pewpan M. Intapan.

**Project administration:** Wanchai Maleewong.

**Software:** Oranat Chuchuen, Thani Thammaratana, Pewpan M. Intapan.

**Writing – original draft:** Oranat Chuchuen, Thani Thammaratana, Oranuch Sanpool, Rutchanee Rodpai, Wanchai Maleewong, Pewpan M. Intapan.

**Writing – review & editing:** Oranat Chuchuen, Wanchai Maleewong, Pewpan M. Intapan.

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
