## [Decision Letter · Decision Letter 0]

1 Aug 2019

PONE-D-19-17696

Rapid label-free identification of Opisthorchis viverrini eggs in fecal specimens using confocal Raman spectroscopy

PLOS ONE

Dear Dr. Maleewong,

Thank you for submitting your manuscript to PLOS ONE. After careful consideration, we feel that it has merit but does not fully meet PLOS ONE’s publication criteria as it currently stands. Therefore, we invite you to submit a revised version of the manuscript that addresses the points raised during the review process.

ACADEMIC EDITOR: The MS by Chuchuen and others presented an interesting alternative for the diagnosis of Opisthorchis eggs, an important task in endemic areas using the microscopy to differentiate from others common trematodes occurring in the same region. However, there are some points to be clarified and important suggestions by the reviewers which are critical to assure the quality of the results and to improve the quality of the study; improving the controls of the experiments, add a blind test comparison by different microscopists, and use samples from human naturally infected. Also, I may suggest a revision on the usage of certain terminology eg.: in the lInes 160-161 it is stated the profile of the eggs are "unique", but there are no tests shown using others helminthes eggs and protozoa oocists to make this statement, also "in situ" or "noninvasive" (line 183 and in other passages) should be carefully checked.

We would appreciate receiving your revised manuscript by Sep 15 2019 11:59PM. To enhance the reproducibility of your results, we recommend that if applicable you deposit your laboratory protocols in protocols.io, where a protocol can be assigned its own identifier (DOI) such that it can be cited independently in the future. For instructions see: http://journals.plos.org/plosone/s/submission-guidelines#loc-laboratory-protocols

We look forward to receiving your revised manuscript.

Kind regards,

Marcello Otake Sato, Ph.D., D.V.M.

Academic Editor

PLOS ONE

2. In your Methods section, please provide additional location information of the collection sites of the fish, including geographic coordinates for the data set if available.

4. To comply with PLOS ONE submissions requirements, please provide methods of sacrifice of the fish in the Methods section of your manuscript.

5. In your Methods section, please include a comment about the state of the hamsters following this research. Were they euthanized or housed for use in further research? If any animals were sacrificed by the authors, please include the method of euthanasia and describe any efforts that were undertaken to reduce animal suffering.

Additional Editor Comments (if provided):

The MS by Chuchuen and others presented an interesting alternative for the diagnosis of Opisthorchis eggs, an important task in endemic areas using the microscopy to differentiate from others common trematodes occurring in the same region. However, there are some points to be clarified and important suggestions by the reviewers which are critical to assure the quality of the results and to improve the quality of the study; improving the controls of the experiments, add a blind test comparison by different microscopists, and use samples from human naturally infected. Also, I may suggest a revision on the usage of certain terminology eg.: in the lInes 160-161 it is stated the profile of the eggs are "unique", but there are no tests shown using others helminthes eggs and protozoa oocists to make this statement, also "in situ" or "noninvasive" (line 183 and in other passages) should be carefully checked.

Reviewers' comments:

Reviewer's Responses to Questions

**Comments to the Author**

1. Is the manuscript technically sound, and do the data support the conclusions?

Reviewer #1: Partly

Reviewer #2: Partly

2. Has the statistical analysis been performed appropriately and rigorously? 

Reviewer #1: N/A

Reviewer #2: N/A

3. Have the authors made all data underlying the findings in their manuscript fully available?

Reviewer #1: Yes

Reviewer #2: Yes

4. Is the manuscript presented in an intelligible fashion and written in standard English?

Reviewer #1: Yes

Reviewer #2: Yes

5. Review Comments to the Author

Reviewer #1: Comments to PONE-D-19-17696 manuscript entitled “Rapid label-free identification of Opisthorchis viverrini eggs in fecal specimens using confocal Raman spectroscopy” by Wanchai Maleewong et al.

Authors’ purposes, in this manuscript, is to “develop a label-free methodology for in situ identification of O. viverrini eggs using non-invasive Raman spectroscopy”. The manuscript seems to me well written, well riding. However, I am not an English native for a good language revision.

Global comments: Authors have choice an interesting, relevant and pertinent diagnosis suggestion. They need to be stimulated to go on working on the field. However, in my point of view, manuscript evidences three main concerns: 1) Lack of information related with the diagnostic value of faecal microscopic examination and its accuracy (specificity and sensitivity; false negatives? etc.); 2) No data related with marker bands for Clonorchis sinensis and other eggs of Opisthorchis’s derived from Raman spectroscopy application; 3) Finally, this manuscript describes results obtained with faecal samples of experimental infection in Hamsters. Why authors do not use faecal human samples? Validate their results with human samples seems to me crucial.

Specific comments: The manuscript describes an innovative and interesting technique. In My Point Of View, now, the manuscript is not in good shape for publication.

Reviewer #2: This manuscript entitled Rapid label-free identification of Opisthorchis viverrini eggs in fecal specimens using confocal Raman spectroscopy presents an originality to apply new technology to solve the problem in differential identification of Opisthorchis viverrini-, Clonorchis sinensis-, O. felineus- and intestinal minutes eggs in fecal sample. This will be rapid and really useful for assessment of actual epidemiology, prevention and control.

However, there are still the major points that need to be addressed by the authors regarding the comments below.

The major concerning

1. Authors need to include other opisthorchiidae eggs, C. sinensis and O. felineus, and species confirm intestinal minutes in this manuscript. As mentioned by the authors that “The averaged Raman spectra of 10 O. viverrini eggs from each infected hamster showed little variability, revealing three unique Raman bands specific to O. viverrini eggs.” (L130-132), it is too early because there is no the spectrum of other opisthorchiidae and intestinal minutes eggs to confirm.

2. Moreover, authors may blind the egg samples and then send to 2-3 microscopic examiners and the ramen spectroscopy for evaluation of reliability, sensitivity and specificity between two techniques.

3. Authors should plan to include eggs from human subject to confirm that it can be used in the human.

6. PLOS authors have the option to publish the peer review history of their article (what does this mean?). If published, this will include your full peer review and any attached files.

Reviewer #1: No

Reviewer #2: No

---

## [Author Response · Author response to Decision Letter 0]

6 Sep 2019

Dear Editor

Thank you for reviewing our manuscript. We appreciate the careful review, comments, and helpful suggestions of you and the reviewers. Please find attached a revised version of our manuscript, “Rapid label-free identification of Opisthorchis viverrini eggs in fecal specimens using confocal Raman spectroscopy,” in submission to PLOS ONE. In it, we have responded, point-by-point, to the comments and suggestions regarding our initial submission, and believe that the manuscript is improved as a consequence. In particular, we have revised the manuscript as follows.

Reply: we have checked and modified the manuscript to meet the style requirements of PLOS ONE.

2. In your Methods section, please provide additional location information of the collection sites of the fish, including geographic coordinates for the data set if available.

Reply: Per your request, we have provided more location information about the collection site in the revised manuscript (lines 71-73).

Reply: The fish used in this study were purchased from a local fish market; thus, no specific permission was required. Please note that we added the sentence, “No specific permission was required for sampling fishes in public locations.” In the Materials and Methods section (line 73).

4. To comply with PLOS ONE submissions requirements, please provide methods of sacrifice of the fish in the Methods section of your manuscript. 

Reply: No humane endpoint was used in this study because we used only dead fish purchased from a local fish market. Please note that we added the sentence, “No humane endpoint was used because only dead fish from a food market were used” (lines77). After obtaining them, the dead fish were processed by the methods outlined in lines 74-81in the Materials and Methods section.

5. In your Methods section, please include a comment about the state of the hamsters following this research. Were they euthanized or housed for use in further research? If any animals were sacrificed by the authors, please include the method of euthanasia and describe any efforts that were undertaken to reduce animal suffering.

Reply: No euthanized and killed animals were used - only the fecal sediment containing O. viverrini eggs was collected. After this study, the hamsters were housed for eggs collection of other studies until the animals are expired. Please see revised manuscript, lines 94-96. Please note that the study protocol was approved by the Animal Ethics Committee of Khon Kaen University (AEMDKKU 029/2018) and animal care was carried out in strict accordance with the recommendations laid out in the National Research Council of Thailand's Guide for the Care and Use of Laboratory Animals. The experiment was made to minimize pain and suffering to the animals and only fecal specimens were collected. Five male hamsters were housed under conventional conditions (12 hours dark-light cycles, temperature 25 ± 2°C), fed with a stock diet (Smart heart, Thailand) and given filtered bottled water ad libitum. The hamster health was observed daily and cage bedding changed twice a week. None of experimental hamsters showed serious illness or severe health problems during the study. 

Additional Editor Comments:

The MS by Chuchuen and others presented an interesting alternative for the diagnosis of Opisthorchis eggs, an important task in endemic areas using the microscopy to differentiate from others common trematodes occurring in the same region. However, there are some points to be clarified and important suggestions by the reviewers which are critical to assure the quality of the results and to improve the quality of the study; improving the controls of the experiments, add a blind test comparison by different microscopists, and use samples from human naturally infected. Also, I may suggest a revision on the usage of certain terminology eg.: in the lInes 160-161 it is stated the profile of the eggs are "unique", but there are no tests shown using others helminthes eggs and protozoa oocists to make this statement, also "in situ" or "noninvasive" (line 183 and in other passages) should be carefully checked.

Reply: The objective of this study was to investigate the feasibility of Raman spectroscopy for rapid, label-free detection of liver fluke eggs in fecal specimens. As a proof-of-concept study, O. viverrini eggs obtained from hamster and human feces were to demonstrate the promising capability eggs liver group of chosen as a reprentative of Raman spectroscopy for this application. Our goal was to report a set of Raman marker bands of O. viverrini eggs that could be used to identify the eggs in feces by the Raman spectroscopic technique. Thus, the control of this study consisted of the Raman spectra of the fecal debris around O. viverrini eggs. Results confirmed that the Raman bands of O. viverrini eggs found were different from those of the surrounding fecal debris (see Fig 5). Clearly, the analysis of others helminthes eggs and protozoa oocysts is very useful and will be an interesting topic for our follow up studies.

This study was conducted in a blind manner. The specimens were observed by two groups of observers. One group consisted of parasitologists who prepared and collected O. viverrini eggs, while the other group consisted of Raman microscopists who ran the Raman spectroscopic experiments and investigated the Raman spectra. 

We appreciate your comments about using human naturally infected O. viverrini eggs in the study. As per your suggestion, we acquired O. viverrini eggs from fecal specimens of four patients and ran additional Raman experiments to obtain Raman spectral data of those human O. viverrini eggs. The results are now presented in the new Figures 3 and 4 and lines 142-146 in the Results section. The peak locations of the Raman spectral bands of human O. viverrini eggs were included in the revised Table 1. Please note that the methods of human specimen acquisition and preparation were included in lines 102 to 109 in the Materials and Methods section. 

As for your additional comments, we have deleted the word “unique” and revised the statement. This revision can be seen in the revised manuscript. In addition, we realized that the use of “in situ” and “noninvasive” might not be suitable for our contexts of work here. We deleted the word “in situ” from the entire manuscript. As for the “noninvasive” word, we realize that we used both “non-invasive” and label-free”, making it too redundant. In fact, “label-free” would be a more suitable word and sufficient to define our technique here. Therefore, we deleted “noninvasive” throughout the manuscript, but have remained to use “label-free” to describe our methodology. 

Reviewer #1: 

Authors’ purposes, in this manuscript, is to “develop a label-free methodology for in situ identification of O. viverrini eggs using non-invasive Raman spectroscopy”. The manuscript seems to me well written, well riding. However, I am not an English native for a good language revision.

Reply: Thank you very much for the positive comments.

Global comments: Authors have choice an interesting, relevant and pertinent diagnosis suggestion. They need to be stimulated to go on working on the field. However, in my point of view, manuscript evidences three main concerns: 

1) Lack of information related with the diagnostic value of faecal microscopic examination and its accuracy (specificity and sensitivity; false negatives? etc.); 

Reply: The objective of this proof-of-concept study was to demonstrate the promising capability of Raman spectroscopy for label-free identification of O. viverrini eggs in hamster and human feces. Clearly, follow-up studies are needed to observe the diagnostic value and accuracy of our Raman spectroscopic technique vs. the traditional faecal microscopic examination for identification of different types of parasitic eggs. 

2) No data related with marker bands for Clonorchis sinensis and other eggs of Opisthorchis’s derived from Raman spectroscopy application

Reply: Our goal was to demonstrate that Raman spectroscopy is a promising technique for detecting liver fluke eggs. Eggs of O. viverrini were chosen as a representative group of liver fluke eggs to develop our technique and demonstrate our proof-of-principle methodology. Clearly, the Raman spectroscopic analysis of Clonorchis sinensis and other eggs of Opisthorchis is very useful, and will be a topic for our follow up analysis. 

3) Finally, this manuscript describes results obtained with faecal samples of experimental infection in Hamsters. Why authors do not use faecal human samples? Validate their results with human samples seems to me crucial.

Reply: We acquired O. viverrini eggs from fecal specimens of 4 human subjects and conducted a new set of additional experiments to acquire Raman spectra of those human-derived O. viverrini egg. The results are now presented in the new Figures 3-4 and the revised Table 1 (lines 142-146). The new results demonstrated close similarity between the Raman spectral bands of O. viverrini eggs derived from hamster and human, confirming that this methodology can be applied to detect O. viverrini eggs in human feces. 

4Specific comments: The manuscript describes an innovative and interesting technique. In My Point Of View, now, the manuscript is not in good shape for publication.

Reply: We appreciate the very helpful comments of the reviewer. We have responded to the suggestions for revisions of our original manuscript, and believe that the revised version of the manuscript has satisfactorily addressed them and, in so doing, has provided a more complete presentation of our study. 

Reviewer #2: 

This manuscript entitled Rapid label-free identification of Opisthorchis viverrini eggs in fecal specimens using confocal Raman spectroscopy presents an originality to apply new technology to solve the problem in differential identification of Opisthorchis viverrini-, Clonorchis sinensis-, O. felineus- and intestinal minutes eggs in fecal sample. This will be rapid and really useful for assessment of actual epidemiology, prevention and control.

However, there are still the major points that need to be addressed by the authors regarding the comments below.

 The major concerning:

1. Authors need to include other opisthorchiidae eggs, C. sinensis and O. felineus, and species confirm intestinal minutes in this manuscript. As mentioned by the authors that “The averaged Raman spectra of 10 O. viverrini eggs from each infected hamster showed little variability, revealing three unique Raman bands specific to O. viverrini eggs.” (L130-132), it is too early because there is no the spectrum of other opisthorchiidae and intestinal minutes eggs to confirm.

Reply: We appreciate that the reviewer pointed this to us. We realized that the original statement was misleading. Therefore, instead of saying that the three Raman bands were unique to O. viverrini eggs, we revised the statement to say that the O. viverrini eggs consisted of three Raman bands. The revised statement can be seen in lines 140-142 in the revised manuscript. We agree that it is too early to conclude that these bands are unique to O. viverrini eggs since there are no data from other eggs shown in this manuscript. Again, we note that our intent for this proof-of-principle study has been to demonstrate application of our new methodology based on Raman spectroscopy by applying it to detect a type of fluke egg, O. viverrini. Currently, we are applying our developed methodology to measure other opisthorchiidae eggs and minute intestinal fluke eggs. The results will be presented in our follow up report. 

2. Moreover, authors may blind the egg samples and then send to 2-3 microscopic examiners and the ramen spectroscopy for evaluation of reliability, sensitivity and specificity between two techniques.

Reply: This study was conducted in a blind fashion in which the specimens were observed by two groups of observers. The first group consisted of parasitologists who prepared O. viverrini eggs, while the second group consisted of Raman microscopists who ran the Raman spectroscopic experiments, examined the Raman spectra, and analyzed the spectral data. 

3. Authors should plan to include eggs from human subject to confirm that it can be used in the human.

Reply: Per the reviewer’s recommendation, we have run a new set of additional experiments to obtain Raman spectral data from human fecal specimens. The results are now included in the revised manuscript in the new Figures 3 and 4. The new results suggested close similarity between the Raman spectral peaks of O. viverrini eggs in fecal specimens of hamster and human, confirming that the methodology can be applied for detection of O. viverrini eggs in the human. 

Again, we appreciate the effort and very helpful feedback and suggestions from the reviewers for improving the manuscript. We believe our revisions have satisfactorily addressed these concerns and that the revised manuscript presents a clearer, more scientifically complete presentation of our methodology. Again, thank you for your time and input. We look forward to your review of this revised manuscript.

Sincerely,

---

## [Decision Letter · Decision Letter 1]

19 Nov 2019

PONE-D-19-17696R1

Rapid label-free identification of Opisthorchis viverrini eggs in fecal specimens using confocal Raman spectroscopy

PLOS ONE

Dear Dr. Maleewong,

Thank you for submitting your manuscript to PLOS ONE. After careful consideration, we feel that it has merit but does not fully meet PLOS ONE’s publication criteria as it currently stands. Therefore, we invite you to submit a revised version of the manuscript that addresses the points raised during the review process.

The revised MS by Chuchuen and others improved several aspects as suggested by the reviewers. However, an important point was addressed by Reviewer 3, which should be adequately addressed.

We would appreciate receiving your revised manuscript by Jan 03 2020 11:59PM. To enhance the reproducibility of your results, we recommend that if applicable you deposit your laboratory protocols in protocols.io, where a protocol can be assigned its own identifier (DOI) such that it can be cited independently in the future. For instructions see: http://journals.plos.org/plosone/s/submission-guidelines#loc-laboratory-protocols

We look forward to receiving your revised manuscript.

Kind regards,

Marcello Otake Sato, Ph.D., D.V.M.

Academic Editor

PLOS ONE

Additional Editor Comments (if provided):

The revised MS by Chuchuen and others improved several aspects as suggested by the reviewers. However, an important point was addressed by Reviewer 3, which should be adequately addressed.

Reviewers' comments:

Reviewer's Responses to Questions

**Comments to the Author**

1. If the authors have adequately addressed your comments raised in a previous round of review and you feel that this manuscript is now acceptable for publication, you may indicate that here to bypass the “Comments to the Author” section, enter your conflict of interest statement in the “Confidential to Editor” section, and submit your "Accept" recommendation.

Reviewer #2: All comments have been addressed

Reviewer #3: (No Response)

2. Is the manuscript technically sound, and do the data support the conclusions?

Reviewer #2: Yes

Reviewer #3: Partly

3. Has the statistical analysis been performed appropriately and rigorously? 

Reviewer #2: N/A

Reviewer #3: N/A

4. Have the authors made all data underlying the findings in their manuscript fully available?

Reviewer #2: Yes

Reviewer #3: Yes

5. Is the manuscript presented in an intelligible fashion and written in standard English?

Reviewer #2: Yes

Reviewer #3: Yes

6. Review Comments to the Author

Reviewer #2: (No Response)

Reviewer #3: I found the aim of the study to be interesting and Raman confocal spectroscopy to be an extremely useful tool to identify Opisthorchis viverrini eggs in fecal samples of patients with human opisthorchiasis. There is a need for a reliable diagnostic tool to solve the diagnosis problems in opisthorchiasis since O. viverrini and small intestinal flukes share the same endemic area and are very similar in size and shape and cannot be clearly differentiated under light microscope. The manuscript is very clear and easy to read, although I am not an English native. The contribution of this work to the diagnosis of Opisthorchis viverrini infection would be essential to prevent adverse outcomes related to O. viverrini infection and even cholangiocarcinoma.

Mayor comments:

This paper is probably publishable as it is if the word “identification” in the title is changed to a more suitable one that does not imply that by performing Raman confocal spectroscopy on fecal eggs would lead to O. viverrini diagnosis. Therefore “Rapid label-free analysis of Opisthorchis viverrini eggs in fecal specimens using confocal Raman spectroscopy” is suggested for the manuscript. Otherwise, mayor revision is recommended to corroborate the identification of the O. viverrini eggs and the following issues should be addressed:

1) The correct diagnosis of Opisthorchis viverrini infection depends on the unequivocally identification of the O. viverrini eggs in fecal samples of the infected patients, which are difficult to differentiate from other parasites with similar morphological features. In order to develop a methodology that overcomes this issue, the authors need to perform Raman spectroscopy of other intestinal flukes that are difficult to differentiate morphologically from those of O. viverrini. Authors should discuss the tentative peak assignments for the main peaks observed in the spectral profile of O. viverrini and the reason they would not expect those peaks to be present in the other intestinal flukes.

2) Given the fact that O. viverrini eggs are difficult to differentiate morphologically from other Opisthorchis-like eggs, it would be of a great value to perform a specific method to validate that authors performed Raman analysis of the O. viverrini eggs.

Minor comments:

1) Authors used normal saline and then centrifugation/filtration to concentrate O. viverrini eggs from fecal samples. Please specify if human fecal samples were fresh or preserved before processing.

2) Line 80, the phrase “Opisthorchis viverrini metacercariae of were examined and identified under a dissecting microscope” should be revised.

7. PLOS authors have the option to publish the peer review history of their article (what does this mean?). If published, this will include your full peer review and any attached files.

Reviewer #2: No

Reviewer #3: No

---

## [Author Response · Author response to Decision Letter 1]

30 Nov 2019

Dear Editor,

Please find attached a revised version of our manuscript, “Rapid label-free analysis of Opisthorchis viverrini eggs in fecal specimens using confocal Raman spectroscopy”, in submission to PLOS ONE. We very much appreciate the careful review and helpful suggestions of the reviewers. In this revised manuscript, we have responded to the comments and suggestions regarding our submission and hope that you find the manuscript improved. Below, we have included our responses to the comments presented by the reviewers. 

Additional Editor Comments (if provided):

The revised MS by Chuchuen and others improved several aspects as suggested by the reviewers. However, an important point was addressed by Reviewer 3, which should be adequately addressed.

Reply: We have revised the manuscript to address the comments raised by Reviewer 3. Please see the revised manuscript and our responses to the reviewer’s comments below.

Reviewer #2: (No Response)

Reviewer #3: I found the aim of the study to be interesting and Raman confocal spectroscopy to be an extremely useful tool to identify Opisthorchis viverrini eggs in fecal samples of patients with human opisthorchiasis. There is a need for a reliable diagnostic tool to solve the diagnosis problems in opisthorchiasis since O. viverrini and small intestinal flukes share the same endemic area and are very similar in size and shape and cannot be clearly differentiated under light microscope. The manuscript is very clear and easy to read, although I am not an English native. The contribution of this work to the diagnosis of Opisthorchis viverrini infection would be essential to prevent adverse outcomes related to O. viverrini infection and even cholangiocarcinoma.

Major comments:

This paper is probably publishable as it is if the word “identification” in the title is changed to a more suitable one that does not imply that by performing Raman confocal spectroscopy on fecal eggs would lead to O. viverrini diagnosis. Therefore “Rapid label-free analysis of Opisthorchis viverrini eggs in fecal specimens using confocal Raman spectroscopy” is suggested for the manuscript. Otherwise, major revision is recommended to corroborate the identification of the O. viverrini eggs and the following issues should be addressed:

1) The correct diagnosis of Opisthorchis viverrini infection depends on the unequivocally identification of the O. viverrini eggs in fecal samples of the infected patients, which are difficult to differentiate from other parasites with similar morphological features. In order to develop a methodology that overcomes this issue, the authors need to perform Raman spectroscopy of other intestinal flukes that are difficult to differentiate morphologically from those of O. viverrini. Authors should discuss the tentative peak assignments for the main peaks observed in the spectral profile of O. viverrini and the reason they would not expect those peaks to be present in the other intestinal flukes.

2) Given the fact that O. viverrini eggs are difficult to differentiate morphologically from other Opisthorchis-like eggs, it would be of a great value to perform a specific method to validate that authors performed Raman analysis of the O. viverrini eggs.

Reply: We highly appreciate the helpful and insightful comments of the reviewer. The reviewer suggested that the word “identification” should be changed. If not changing it, then we should “corroborate the identification of the O. viverrini eggs” by addressing the two additional issues raised. We fully agree with the reviewer’s opinion that the word “identification” is misleading and not really suitable to describe our technique presented here. Therefore, as suggested by the reviewer, we changed the word “identification” to other more appropriate words in order to provide a clearer, more accurate representation of our study. In the title we changed the word “identification” to “analysis”, as suggested by the reviewer. In addition, the word “identification” was replaced by other more suitable words throughout the manuscript, as follows:

• The title was changed to: “Rapid label-free analysis of Opisthorchis viverrini eggs in fecal specimens using confocal Raman spectroscopy”. 

• The short title was changed to: “Analysis of Opisthorchis viverrini eggs using Raman spectroscopy”. 

• In the Abstract section, the words “identification” (lines 27 and 29) and identified (line 28) were changed to “analysis” and “reported”, respectively. 

• In the Introduction section, the words “identification” (lines 47, 61, and 64) and “identified” (line 63) were changed to “analysis” and “reported”, respectively.

• The caption of Fig 1,“Experimental setup for identification of Opisthorchis viverrini eggs in fecal specimens” was changed to “Experimental setup for confocal Raman analysis of Opisthorchis viverrini eggs in fecal specimens”.

• In the Results section, the word “identification” (line 184) was changed to “ label-free Raman analysis”. 

• In the Discussion section, the word “identification” (lines 205 and 212) was changed to “analysis”. Also, in line 215, the phrase “to rapidly and precisely identify O. viverrini eggs” was changed to “to rapidly examine O. viverrini eggs”.

• In the Discussion, the statement “the proof-of-principle methodology developed here could enable the precise identification and differentiation of different types of fluke eggs in a label-free manner” (lines 221-222) was changed to “the proof-of-principle methodology developed here could enable the analysis of different types of fluke eggs in a label-free manner”

Again, we appreciate the very helpful and thoughtful suggestions from the reviewer. Our intention for this proof-of-principle study has been to demonstrate application of Raman spectroscopy to provide a label-free analysis of O. viverrini eggs. Currently, we are applying our developed methodology to measure other opisthorchiidae eggs and minute intestinal fluke eggs. The results will be presented in our follow up report.

Minor comments: 

1) Authors used normal saline and then centrifugation/filtration to concentrate O. viverrini eggs from fecal samples. Please specify if human fecal samples were fresh or preserved before processing.

Reply: The human fecal samples were fresh before processing. As per the reviewer’s request, we have specified it in the text by adding a new sentence (lines 104-106): “Fresh human fecal specimens were processed using the methods for processing hamster fecal samples, as outlined above.” 

2) Line 80, the phrase “Opisthorchis viverrini metacercariae of were examined and identified under a dissecting microscope” should be revised.

Reply: We apologize that our original statement was incomplete. To correct the statement and improve its clarity, we changed it to “ The collected Opisthorchis viverrini metacercariae were examined and identified under a dissecting microscope, as previously described [1].” (lines 78-80).

Again, we appreciate the effort on the part of the reviewers in providing thoughtful feedback and suggestions for improving the manuscript. We believe our revisions have satisfactorily addressed them and that the revised manuscript provides a clearer, more complete presentation of our methodology. Again, thank you for your time and input. We look forward to your review of this revised manuscript.

Yours sincerely,

The authors

---

## [Decision Letter · Decision Letter 2]

6 Dec 2019

Rapid label-free analysis of Opisthorchis viverrini eggs in fecal specimens using confocal Raman spectroscopy

PONE-D-19-17696R2

Dear Dr. Maleewong,

We are pleased to inform you that your manuscript has been judged scientifically suitable for publication and will be formally accepted for publication once it complies with all outstanding technical requirements.

With kind regards,

Marcello Otake Sato, Ph.D., D.V.M.

Academic Editor

PLOS ONE

Additional Editor Comments (optional):

The authors addressed all the suggestions of the reviewers improving consistently the MS.

Reviewers' comments:

Reviewer's Responses to Questions

**Comments to the Author**

1. If the authors have adequately addressed your comments raised in a previous round of review and you feel that this manuscript is now acceptable for publication, you may indicate that here to bypass the “Comments to the Author” section, enter your conflict of interest statement in the “Confidential to Editor” section, and submit your "Accept" recommendation.

Reviewer #3: All comments have been addressed

2. Is the manuscript technically sound, and do the data support the conclusions?

Reviewer #3: Yes

3. Has the statistical analysis been performed appropriately and rigorously? 

Reviewer #3: N/A

4. Have the authors made all data underlying the findings in their manuscript fully available?

Reviewer #3: Yes

5. Is the manuscript presented in an intelligible fashion and written in standard English?

Reviewer #3: Yes

6. Review Comments to the Author

Reviewer #3: 1) Reference Zheng et al. (2017), pages must be corrected to 11-19.

2) Reference Crow et al. (2003), pages must be corrected to 106-8.

3) Please unify criteria about including the issue number.

7. PLOS authors have the option to publish the peer review history of their article (what does this mean?). If published, this will include your full peer review and any attached files.

Reviewer #3: No

---

## [Editor Report · Acceptance letter]

11 Dec 2019

PONE-D-19-17696R2 

Rapid label-free analysis of *Opisthorchis viverrini* eggs in fecal specimens using confocal Raman spectroscopy 

Dear Dr. Maleewong:

I am pleased to inform you that your manuscript has been deemed suitable for publication in PLOS ONE. Congratulations! Your manuscript is now with our production department. 

With kind regards,

on behalf of

Dr. Marcello Otake Sato 

Academic Editor

PLOS ONE